# Acceleration of Bone Fracture Healing through the Use of Bovine Hydroxyapatite or Calcium Lactate Oral and Implant Bovine Hydroxyapatite–Gelatin on Bone Defect Animal Model

**DOI:** 10.3390/polym14224812

**Published:** 2022-11-09

**Authors:** Aniek Setiya Budiatin, Junaidi Khotib, Samirah Samirah, Chrismawan Ardianto, Maria Apriliani Gani, Bulan Rhea Kaulika Hadinar Putri, Huzaifah Arofik, Rizka Nanda Sadiwa, Indri Lestari, Yusuf Alif Pratama, Erreza Rahadiansyah, Imam Susilo

**Affiliations:** 1Department of Pharmacy Practice, Faculty of Pharmacy, Universitas Airlangga, Surabaya 60115, Indonesia; 2Department of Orthopaedics and Traumatology, Faculty of Medicine, Universitas Airlangga, Surabaya 60131, Indonesia; 3Department of Anatomical Pathology, Faculty of Medicine, Universitas Airlangga, Surabaya 60131, Indonesia

**Keywords:** defect, bone remodeling, bovine hydroxyapatite, calcium lactate, BHA–GEL pellet

## Abstract

Bone grafts a commonly used therapeutic technique for the reconstruction and facilitation of bone regeneration due to fractures. BHA–GEL (bovine hydroxyapatite–gelatin) pellet implants have been shown to be able accelerate the process of bone repair by looking at the percentage of new bone, and the contact between the composite and bone. Based on these results, a study was conducted by placing BHA–GEL (9:1) pellet implants in rabbit femoral bone defects, accompanied by 500 mg oral supplement of BHA or calcium lactate to determine the effectiveness of addition supplements. The research model used was a burr hole defect model with a diameter of 4.2 mm in the cortical part of the rabbit femur. On the 7th, 14th and 28th days after treatment, a total of 48 New Zealand rabbits were divided into four groups, namely defect (control), implant, implant + oral BHA, and implant + oral calcium lactate. Animal tests were terminated and evaluated based on X-ray radiology results, *Hematoxylin-Eosin* staining, vascular endothelial growth Factor (VEGF), osteocalcin, and enzyme-linked immunosorbent assay (ELISA) for bone alkaline phosphatase (BALP) and calcium levels. From this research can be concluded that Oral BHA supplementation with BHA–GEL pellet implants showed faster healing of bone defects compared to oral calcium lactate with BHA–GEL pellet implants.

## 1. Introduction

Bone is a special connective tissue that hardens via the process of mineralization by calcium phosphate in the form of hydroxyapatite [1]. Various kinds of bone, joint, and muscle diseases in humans include open fractures, closed fractures, osteoporosis, osteoarthritis, osteomalacia, osteomyelitis, rheumatic polymyalgia, gouty arthritis, rheumatoid arthritis and others, with fractures being the most common large organ traumatic injury in humans [2]. A fracture is neuromuscular damage due to trauma to the tissue [3] and results in a gap in the bone. Fracture repair can generally restore damaged skeletal organs to their preinjury cellular composition, structure, and biomechanical function, but about 10% of fractures will not heal normally [2]. In 2011, the World Health Organization (WHO) recorded more than 1.3 million people suffering from fractures due to accidents. Accident cases that have a fairly high prevalence, amongst which are lower extremity fractures, reprsent 40% of the accidents that occur. Bone graft therapy, with a surgical procedure that places new bone or replacement material (composite matrix) into the space around the fracture or hole in the damaged bone (defect) to help speed up the healing process [3], is commonly used in fracture management [4].

Bovine hydroxyapatite (BHA) is an inorganic bovine bone material used as an alternative composite component, consisting of 93% hydroxyapatite (Ca_10_(PO_4_)_6_(OH)_2_) and 7% β-tricalcium phosphate (Ca_3_(PO_4_)_2_). As a result, it is more porous and can only absorb antibiotics, hormones or growth factors [5]. BHA has properties similar to hydroxyapatite in human bone and is a scaffold that is more osteoconductive than other synthetic hydroxyapatites, is biocompatible and has high porosity. The high porosity of BHA accelerates the process of colonization of osteoblast cells and becomes a medium for osteoblast cells to stick to [6]. On the other hand, BHA is brittle as a new bone-forming material, so gelatin is added as an adhesive and smoothing agent [5]. Gelatin (GEL) is a macromolecule produced by partial hydrolysis of collagen from skin, white connective tissue and animal bones of amino acid residues [7]. GEL is commonly called type one collagen which together with osteoblasts forms osteoids (soft callus). BHA–GEL composites that resemble mineral components in humans are able to form new bone and fill bone gaps due to fractures [5] with high biocompatible properties, osteoconductive, osteoinductive, biodegradable, bioresorbable, and non-toxic [8,9]. The addition of gelatin can also control the degradation of pellet time and increase the synthesis of new bone in the defect area [10].

BHA–GEL composite implants have been shown to be able to accelerate the bone repair process in fractured rabbit femurs within 28 to 42 days, and have good biomaterials for bone filling [11]. However, this period of time is still relatively long when considering the effect of pain felt by patients with fractured bones, and can affect the patient’s psychological (anxiety) level [12]. Repair of bone mineralization alone is not sufficient to meet demand and can affect skeletons with inadequate bone tissue function. It is necessary to increase the supply of calcium. Simple calcium preparations are generally administered orally as an adjuvant to treatment, in combination with more specific drugs [13]. For this reason, a study was carried out on bone defects due to fractures with BHA–GEL pellet implants and the addition of BHA or calcium lactate orally as calcium supplements. Calcium intake plays an important role in maintaining bone health, namely to achieve peak bone mass and prevent loss of bone mass [14]. The purpose of supplementation in this study was to prove that the period of bone growth around the defect could be shortened. Osteoblast proliferation is expected to increase with the addition of oral BHA supplements to form more osteoids (soft callus), which are then converted into osteocytes (hard callus) and can increase bone stability around the fracture.

BHA is known to have carbonate substitutions like that of human hydroxyapatite and that can be found in synthetic biomimetic hydroxyapatites [15,16]. The carbonate group increases the proliferation of osteoblasts, thereby accelerating the formation of new bone [17]. Hydroxyapatite raises blood calcium levels less than calcium carbonate and calcium citrate. This indicates that hydroxyapatite is more effective at entering bone cells [18]. BHA bioavailability is better than calcium carbonate [14] and absorbed about 42.5% [19]. A European study showed that hydroxyapatite was more effective than calcium carbonate in slowing bone loss of the peripheral trabeculae of the distal tibia and distal radius [14]. The role of oral BHA is to increase the proliferation of osteoblasts by stimulating mesenchymal cells. It is shown by the calcium deposition from hydroxyapatite will interact with the collagen fibers along with type I collagen, a substance from the degradation of gelatin. During the mineralization process, the ends of the bone fragments are covered by a fusiform mass filled with woven bone. The more minerals deposited, the harder the callus formed [20]. While calcium lactate which is a salt form of lactic acid in the form of white powder, crystals or grains [21], which is also given orally because it can increase the number of osteoblasts [22]. Calcium lactate intake can increase extracellular calcium and intracellular calcium levels, stimulating bone formation significantly and increasing the proliferation or chemotaxis of osteoblasts [22]. Calcium lactate also significantly reduces bone resorption [23]. In addition, calcium lactate in the body is easily converted into calcium bicarbonate, and only about 25% is absorbed in the small intestine from calcium intake through passive diffusion and active transport [24]. A study shows a comparison of 500 mg calcium lactate, 500 mg carbonate and 500 mg gluconate (it is known that each calcium content is different), the results of which show the absorption, AUC and excretion of calcium lactate to be better than others [25]. Calcium lactate absorption is better than calcium phosphate and calcium in milk, and stimulates bone activity more than calcium carbonate or calcium citrate in experimental rats. As a consequence, it is very effective, especially in bone metabolism [22]. However, based on the research of previous results, it is known that BHA has more osteoinductive properties that other materials do not have [26]. For this reason, it is possible that oral administration of BHA on BHA–GEL pellet implants is more effective in terms of bone growth than oral administration of calcium lactate on BHA–GEL pellet implants.

## 2. Materials and Methods

### 2.1. Ethical Approval

The submission of an ethical feasibility proposal was addressed to the Research Ethics Commission of the Faculty of Veterinary Medicine, Universitas Airlangga (Animal Care and Use Committee/ACUC) and has been declared ethically eligible via Ethical Clearence No. 2.KEH.075.05.2022.

The research was conducted in the laboratory of the Faculty of Pharmacy, Universitas Airlangga, Surabaya in a true experimental manner with a posttest-only control group study design using 48 New Zealand rabbits aged 4–8 months, weighing 1.5–2.5 kg, healthy and without bone disorders in femur. The rabbits were randomly divided into defect (control groups), BHA–GEL pellet implant group, BHA–GEL pellet implant group with oral BHA and BHA–GEL pellet implant group with oral calcium lactate. Rabbits were adapted for one week with adequate food provided during the study. Making a rabbit fracture model was made by generating a defect in the femur, followed by implanting a BHA–GEL pellet implant according to the group division. Furthermore, oral supplements of BHA or calcium lactate were administered until termination was carried out on days 7, 14 and 28, according to group division. After termination, the femur bone that was treated as a sample was taken; this was followed by evaluation through X-ray radiology, and then bone decalcification, to evaluate the number and distribution of bone cells through HE (*Hematoxylin-Eosin*) staining, evaluation of anti-vascular endothelial growth factor (VEGF) and osteocalcin levels using the immunohistochemical (IHC) method.

### 2.2. Materials

In terms of test materials, bovine hydroxyapatite (Universitas Airlangga, Indonesia), gelatin, sodium carboxymethyl cellulose powder, aquadest, ketamine, xylazine, gentamicin ointment, ampicillin, 70% alcohol (pharmaceutical grade), cotton balls, povidone iodine, savlon, cotton bud, sterile gauze, hypafix, handsaplast, calcium lactate 500 mg tablet, thrombophob gel, water for injection, 10% formalin buffer, and a Calcium Colorimetric Assay Kit, ELISA Kit (Cat No. MAK022) (Sigma-aldrich, St. Louis, MO, USA) were acquired. In terms of equipment, a Carver manual pellet press, punch and die (4 mm diameter), 1 cc and 3 cc syringe, bone drill with 4.2 mm drill bit, surgical blade, forceps, needle holder, needle circle or surgical needles, Silk no.3 surgical thread, gillette razor, shaver, tweezers, scissors, leukoplast, vacutainer gel separator, water bath, mortar, stamper, granule sieve, oven, feeding tube, oral catheter, X-ray machine, ELISA reader, light microscope, histology slides, object glass and cover glass were used.

### 2.3. BHA–GEL Pellet Preparation

Weigh and put 9 g Bovine hydroxyapatite (BHA) powder into a mortar and then reduce the particle size using a stamper. Heat 6 mL of distilled water in a glass beaker at 40 °C, then add 2 g of gelatin and stir until the gelatin dissolves. Put 3 mL of dissolved gelatin into a mortar containing BHA and stir until the mass is formed. Next, sieve the mass to obtain a uniform particle size and dry in an oven at 40 °C for 24 h. Weigh the granules as large as 100 mg and compress with a load of 1 ton with a diameter of 4 mm to form BHA–GEL pellets, and continue with UV sterilization for ±3 h.

Implantation of BHA–GEL implant and oral administration of BHA or oral calcium lactate is performed by injecting a combination anesthetic ketamine 50 mg/kgBW and xylazine 5 mg/kgBW intramuscularly in experimental animals. Clean the rabbit’s thighs with 70% alcohol and shave. Disinfect the shaved area using betadine then make an incision in the required area of about 1.5 cm. The defect was made using a 4.2 mm drill bit and followed by implantation of BHA–GEL pellets. Next, suture the wound, disinfect it with 70% alcohol and then betadine, and then apply gentamicin ointment as an antibiotic. Cover the wound with sterile gauze and dressing retention tape, then administer injection of Ampicillin intramuscularly at a dose of 25 mg/kgBW as an antibiotic. During the recovery period, the wound was treated with betadine and tape replacement until the surgical wound was dry. In addition, the rabbits were treated with 1 mL oral BHA or calcium lactate according to the group division.

### 2.4. Characterizations of Pellet

The characterization of the prepared pellets was carried out using Fourier Transform Infrared (FT-IR) Spectroscopy (Perkin Elmer, MA, USA). The BHA, Gelatin, and BHA–GELatin that have been made were then mixed with potassium bromide to make pellets and measured at a wave number of 400–4000 cm^−1^ with only one scanning, while the size and morphology of the particles were observed using scanning electron microscopy (Inspect S-50, FEI, Japan). The sputter coating of SEM used the ultra-thin coating of gold.

### 2.5. Blood Sampling Technique

Xylol was applied to the rabbit’s ear on the marginal vein and 3 mL of blood were using a disposable syringe one hour from the time of taking the drug; this was carried out on the 7th, 14th, and 28th days before termination. Applying heparin gel to the area around the injection to prevent blood clots. Inject blood into a vacutainer containing a gel separator (serum separator tube) (OneMed, Krian, Indonesia) slowly to prevent hemolysis. Continue to centrifuge at 4000 rpm for 15 min to obtain a serum. Serum was stored in the freezer at −80 °C. BALP levels were using ELISA microplate reader IMark series No. 12096 (Bio-Rad, Hercules, CA, USA), while calcium levels using a microplate reader Biochrom EZ Read 2000 serial num-ber 135247 (Biochrom Ltd, Cambridge, United Kingdom.

### 2.6. Bone Sampling Technique

Termination of the experimental rabbits on the 7th, 14th and 28th days after treatment with oral drugs was performed in the instance of both BHA and calcium lactate. A bone sample of the femur in a 10% formalin was followed followed by observation of the process of closing the bone gap by X-ray radiology. Then decalcification of bone in 10% EDTA solution for observation of bone cell development through HE staining (*Hematoxylin-Eosin*), examination of VEGF and osteocalcin levels using the immunohistochemical (IHC) method.

### 2.7. Radiology Examination

An evaluation of bone integrity was performed using X-ray radiography and clarified with *ImageJ V1.44p*, before then compared being with the initial diameter of the bone defect (4.2 mm). This was followed by macroscopic observations to see the percentage of callus growth around the bone defect and calculations using Lane–Sandhu Scoring (Table 1).

### 2.8. Haematoxylin and Eosin Staining

The histological examination started with processing the paraffin blocks by dehydrating with alcohol concentration 70% to 100% for 60 min each. This was followed by 3 clarifications of xylene, for 15 min each time. After that, in an incubator at 60 °C, the permeation treatment with paraffin solution was carried out three times for 60 min each. The tissue was then immersed in liquid paraffin and brought to room temperature. Each paraffin block was then cut to a thickness of 4–6 m using a microtome. Cell morphology was determined with *hematoxylin* and *eosin*. The slides were dipped in xylene three times for 5 min each and hydrated with alcohol (96% to 70% alcohol) for 2 min each. The slides were then rinsed under running water for 10 min, placed in *Mayer’s hematoxylin* for 15 min, rinsed with running water and examined microscopically. The slides were then placed in 1% eosin solution for 30 s, dried, washed, and mounted with an EZ mount. This was performed on five visual fields, with 400× magnification around the defect or implant area. The results obtained are the number of each bone cell, including osteoblasts, osteoclasts and osteocytes.

### 2.9. Immunohistochemistry

The immunohistochemical technique was used to stain VEGF and osteocalcin immunopositive cells. Slides that have been paraffinized are soaked with an antigen retrieval decloaking chamber, cooled for 20 min, and washed with PBS for 3 min. Sniper blocking followed for 15 min. The slides were then incubated with rabbit anti-rat VEGF primer (cat. no. PA1-21796, Thermo Fisher Scientific, 1:100 dilution) (Waltham, Massachusetts, United States) and washed in PBS for 3 min. After that, the universal link was performed for 20 min, and the slide was washed in PBS for 3 min. The Trecavidin-HRP Label was then applied for 10 min and washed with PBS for 3 min. The slides were then reacted with Chromogen DAB + Buffer Substrate for 2–5 min, followed by rinsing for 5 min with running water. The slides were then stained with *hematoxylin* for 1–2 min followed by rinsing for 5 min with running water (twice). The slides were then dehydrated with alcohol (70% absolute alcohol) for 5 min each, followed by three xylol washes for 5 min each. Finally, they were mounted on a slide (Ecomount) and covered with a cover glass.

### 2.10. Bone Alkaline Phosphatase

Examination of BALP levels was carried out using the Rabbit Bone-Specific Alkaline Phosphatase ELISA Kit (Cat. No. BZ-08173140-EB) (Bioenzy, Jakarta, Indonesia) with the following steps: prepare all reagents, standard solutions and samples; bring all reagents to room temperature before use. Add 50 μL standard to standard well; add 40 μL sample to sample wells and add 10 μL anti-BAP antibody to sample wells; then add 50 μL streptavidin-HRP to sample wells and standard wells (not blank control well) and mix well. Cover the plate with a sealer. Incubate for 60 min at 37 °C. After that, remove the sealer and wash the plate 5 times with a wash buffer. Soak wells with at least 0.35 mL wash buffer for 30 s to 1 min for each wash. Blot the plate onto paper towels or other absorbent material. Then, add 50 μL substrate solution A to each well and then add 50 μL substrate solution B to each well. Incubate plate covered with a new sealer for 10 min at 37 °C in the dark. Add 50 μL Stop Solution to each well, and the blue color will change into yellow immediately. Determine the optical density (OD value) of each well immediately, using a microplate reader set to 450 nm within 10 min after adding the stop solution.

### 2.11. Calcium Concentration

Examination of calcium levels in the blood is carried out using the Calcium Colorimetric Assay Kit (Cat. No. MAK022) (Sigma-aldrich, St. Louis, MO, USA). Serum samples can be used directly in this assay. Add 90 mL of the Chromogenic Reagent (Sigma-aldrich, St. Louis, MO, USA) to each well containing standards, samples, or controls and mix gently. Then, add 60 mL of Calcium Assay Buffer (Sigma-aldrich, St. Louis, MO, USA) to each well and mix gently. After that, incubate the reaction for 5–10 min at room temperature. Protect the plate from light during incubation. Measure the absorbance at 575 nm (A575) before assay.

### 2.12. Statistical Analysis

Evaluation of the size of the bone gap that was closed radiologically was analyzed used the Shapiro–Wilk test to determine the normality of the data and homogeneity test using the Levene test. If the data is normally distributed, then the one-way ANOVA test is continued. If the results show a difference in meaning, then it is continued with the LSD post hoc test. Meanwhile, if the data is not normally distributed; then, the Kruskal–Wallis test is carried out and then the Mann–Whitney test. This also applies to the evaluation of number of osteoblasts, osteocytes, and osteoclasts by HE staining, BALP and calcium levels in the blood. The evaluation percentage of callus growth around bone defects was assessed by means of Lane–Sandhu scoring. Evaluation of VEGF and osteocalcin expression using immunohistochemistry was performed semi-quantitatively via the Remmele method. The three tests were analyzed using the Kruskal–Wallis test, and assessment was continued with the Mann–Whitney test if there were significant differences between groups.

## 3. Results

### 3.1. Characterizations of Pellet

The results of the SEM identification show that the resulting pellet has a hexagonal particle shape with a particle size mean of 1.350 ± 0.243 μm (Mean ± SD) (Figure 1a). The results of the FT-IR from BHA describe the percentage of transmission at the specific wave numbers PO_4_ = 1048.57 cm^−1^, OH = 3571.03 cm^−1^, and CO_3_ = 1460.65 cm^−1^, which are the harsh characteristics of bovine hydroxyapatite. Meanwhile, the result of the assessment in gelatin describe the emergence of the percentage of transmission in OH; NH_2_; = 3500–3000 cm^−1^; C=O and NH = 1655–1540 cm^−1^; COO = 1450–1240 cm^−1^ (Figure 1b–d).

### 3.2. Radiology Examination

A study on the effect of oral BHA or calcium lactate on the repair of bone defects, implanted with BHA–GEL pellets, was carried out using a burr hole defect model in the cortical bone of the rabbit femur. Defects with implants appear circular, with higher intensity than their surroundings. The radiographic results showed that the implant group experienced accelerated bone growth in the area of the defect compared to the group without implants. The pellet intensity in the implant + oral BHA group was known to be fainter than that in the implant + oral calcium lactate group (Figure 2a,b), indicating the union of bone with the composite. The results of the evaluation of bone gap closure-obtained data that were not normally distributed (*p* < 0.05) nor homogeneous (*p* < 0.05) in the termination group on days 7, 14 and 28. Furthermore, using the Kruskal–Wallis, it was found that there was no difference which was significant (*p* > 0.05) between groups for each termination period (Figure 2c). The development of bone regeneration can also be known based on callus formation, which is characterized by changes in the intensity and transformation of the implants implanted. The longer treatment times showed the intensity of the pellets was fading, but the radiological results did not describe the percentage of callus growth so that it was continued with macroscopic observations. Lane–Sandhu scoring results (macroscopic observation) showed that the longer the treatment time, the higher the percentage of callus growth. The data generated from the scoring process includes non-parametric data so that the test carried out is Kruskal–Wallis and shows a sig. value (*p* > 0.05), or there is no significant difference between groups in each termination period (Figure 2d).

### 3.3. Examination of the number of Osteoblasts, Osteoclasts and Osteocytes through Hematoxylin-Eosin Staining

The data from the observation of osteoblast cells (Figure 3a) obtained showed that the data were normally distributed (*p* > 0.05) and homogeneous (*p* > 0.05) for each termination period. On days 7 and 28, there was no significant difference (one-way ANOVA; *p* > 0.05) between groups. Meanwhile, on the 14th day, it was known that the implant + oral BHA and implant + oral calcium lactate groups were significantly different (one-way ANOVA post hoc LSD; *p* < 0.05) compared to the defect group (control) (Figure 3b).

The data from the observation of osteoclast cells (Figure 3a) on days 7 and 28 were normally distributed (*p* > 0.05) and homogeneous (*p* > 0.05), and there was no significant difference (one-way ANOVA; *p* > 0.05) between groups. On day 14 the data were not normally distributed (*p* < 0.05) and there was no significant difference (Kruskal–Wallis; *p* > 0.05) between groups (Figure 3c).

The data from the observation of osteocyte cells (Figure 3a) obtained data that were normally distributed (*p* > 0.05) and homogeneous (*p* > 0.05) for each termination period. On days 7 and 14 there was no significant difference (one-way ANOVA; *p* > 0.05) between groups. Meanwhile, on day 28, it was found that the implant + oral BHA group was significantly different from the defect group (control), the implant group and the implant + oral calcium lactate group (one-way ANOVA post hoc LSD; *p* < 0.05). This indicates the effect of the addition of oral BHA on the number of osteocyte cells (Figure 3d).

### 3.4. Examination of the Amount and Distribution of VEGF through Immunohistochemistry

Immunohistochemical examination with anti-VEGF was performed to determine the vascularity of bone tissue in each treatment group. The results of the observation of VEGF expression using the Kruskal–Wallis test analysis in the negative control group, the implant group, the implant + oral BHA group, and the implant + oral calcium lactate group at 7, 14, and 28 days (Figure 4b) showed different values. VEGF expression was calculated (*p* < 0.05). When the results of VEGF expression in the implant and oral BHA group and the implant and oral calcium lactate group on days 7, 14, and 28, there was no difference in VEGF expression value (*p* > 0.05) with the Mann–Whitney test.

### 3.5. Examination of the Amount and Distribution of Osteocalcin through Immunohistochemistry

Immuno-histochemical examination with anti-osteocalcin was performed to determine the number of osteoblasts which are markers of mineral deposition and growth of mature callus in each treatment group. The results of observations of osteocalcin expression in the negative control group, implant group, implant + oral BHA group, and implant + oral calcium lactate group at 7, 14, and 28 days (Figure 5b) showed differences in the value of osteocalcin expression (*p* < 0.05) with the Kruskal–Wallis test. On days 14 and 28, the implant + oral BHA group showed higher IRS scores than the implant + calcium lactate group (Mann–Whitney; *p* < 0.05).

### 3.6. Examination of BALP Levels through ELISA

The results of measurements of BALP levels obtained data that were normally distributed (*p* > 0.05) and homogeneous (*p* > 0.05). There was a significant difference between the implant + oral BHA group and the implant + oral calcium lactate compared to the defect and implant groups on day 7. Meanwhile, on day 14, there was a significant difference between the defect group with an implant, an implant + oral BHA, and an implant + oral calcium lactate (One-way ANOVA post hoc LSD; *p* < 0.05) (Figure 6a).

### 3.7. Examination of Calcium Levels in the Blood through the Calcium Colorimetric Assay Kit

The results of the measurement of calcium levels in the blood obtained data that were normally distributed (*p* > 0.05) and homogeneous (*p* > 0.05). There was a significant difference between the defect group with implant + oral BHA, implant + oral calcium lactate on day 14. Meanwhile, on day 28 there was a difference between the defect group with implant and implant + oral calcium lactate (one-way ANOVA post hoc LSD; *p* < 0.05) (Figure 6b).

## 4. Discussion

An in vivo study to test the effectiveness of oral BHA and calcium lactate in conjunction with BHA–GEL implants was performed on femoral bone defects in rabbits. Based on previous studies, BHA–GEL implants showed accelerated bone growth in terms of the percentage of new bone and bone-composite contact [10]. Castelo et al. [28] stated that the ossein hydroxyapatite complex (OHC) consisting of ossein, a protein that forms the organic matrix of bone and hydroxyapatite (Ca_5_[PO_4_]_3_OH) has been shown to be effective in maintaining bone mineral density (BMD) and has a strong osteogenic effect which is stronger than calcium supplements. Other studies have shown that OHC stimulates bone metabolism by stimulating osteoblast differentiation, activity, and proliferation [29]. OHC has been shown to reduce the rate of bone resorption and stimulate ossification. Administration of OHC as monotherapy for 1–4 weeks in fracture patients has shown a reduction in the time required for healing by stimulating callus formation, thereby accelerating consolidation and clinical improvement [30].

From the characterizations of the pellet, the morphological shape of the particle was hexagonal, with a particle size mean of 1.350 ± 0.243 μm (mean ± SD). This is in line with previous studies of natural hydroxyapatite, one of which shows BHA possess a hexagonal particle shape [31,32]. Several factors that can influence the morphology of the particles are changes in the calcination temperature, as described by Khoo et al. [33]. Then, the FTIR spectra showed the specific wave number of functional group of the structure as well as the BHA–GEL pellet. The band at 1048 cm^−1^ is attributed to an asymmetric stretching vibration mode ν_3_ of PO_4_^3-^, while two sharp peaks at 570 nd 602 cm^−1^ are attributed to asymmetric bending vibration mode ν_3_, and symmetric bending vibration mode ν_2_ of PO_4_^3−^ group. The bands observed at 962 and 1458 cm^−1^ at the FTIR spectra indicate the presence of B-type carbonate CO_3_^2−^ bands [34].

Radiological results revealed the development of bone growth; after measuring bone gap closure, it was found that the implant + oral BHA group (the average diameter on day 28 is 1.910 mm) had a smaller final diameter of the bone gap than the implant + oral calcium lactate group (the average diameter on day 28 is 2.487 mm). This is supported by the observation of callus growth in bone defects, where changes in transformation and pellet intensity were more faded in the implant + oral BHA group. This indicated the growth of new tissue that was fused (union) with the bone due to the penetration of tissue cells around the bone into the implant.

The reparative process of bone fractures involves a series of events that include migration, proliferation, differentiation and activation of several cell types. These include mesenchymal and hematopoietic stem cells that ultimately lead to bone formation and remodeling [35]. These stem cells will differentiate into several types of bone cells, including osteoblasts and osteoclasts, while osteocytes are transformed osteoblasts that are embedded in the bone matrix. All three cells were expressed by *Hematoxylin-Eosin* staining. On day 14, it was found that the osteoblast cells in the implant + oral BHA group were significantly different from the defect group (control). In accordance with the previous theory, where BHA has a carbonate group which is known to increase the proliferation and differentiation of osteoblasts as well as mesenchymal stem cells which are osteoprogenitor cells that produce osteoblasts, thus bone matrix synthesis occurs more quickly [36]. Furthermore, there was also a significant difference between the implant + oral calcium lactate group and the defect group (control). In previous studies, it was stated that calcium lactate can increase extracellular calcium and intracellular calcium levels, thereby stimulating bone formation significantly, as well as increasing osteoblast proliferation or chemotaxis [22]. The same thing was observed in the observation of the number of osteocytes, where an increase in osteoblast proliferation would result in more mineralized osteoid so that the number of osteocytes also increased.

Osteoclasts on day 14 of the defect group (control) increased, indicating that the activity of mature osteoclasts was maximal in resorption of bone in the second week. However, in the group with the implant, the number of osteoclasts was lower than in the defect group (control). This may have been because the group with the implant treatment or the addition of calcium supplements had accelerated fracture healing. Substitution of carbonate contained in BHA causes bioresorption to occur earlier [37]. Increased osteoclastogenesis accelerates cartilage resorption and increased osteoblastogenesis during fracture healing promotes bone fusion, whereas inhibition of osteoclast or osteoblast differentiation has been reported to delay bone healing [38]. The processes of resorption by osteoclasts and bone formation by osteoblasts proceed sequentially but overlap substantially, so that there is a change in the cell population that indicates regeneration in the tissue [2]. This can be seen through the results of research where when there is an increase in the number of osteoblasts, the number of osteoclasts decreases, or vice versa.

Parameters measured by the IHC method were vascular endothelial growth factor (VEGF) and osteocalcin. VEGF is a protein that plays an important role in endothelial proliferation, migration, and activation [39]. VEGF can be detected in the first week after injury, namely in the inflammatory phase [40]. The results of the examination of VEGF expression in the implant + oral BHA group compared to the implant + oral calcium lactate group on days 7, 14, and 28 showed no significant difference. This is probably because VEGF expression in the implant + oral BHA group was seen on day 3, and then peaked on day 5 [41]. In the implant + oral BHA group, there may have been a decrease in the number of macrophages, so that VEGF expression decreased. This decrease was influenced by a decrease in the number of blood vessels damaged in the wound; the extracellular matrix begins to fill the missing area and stable blood vessels begin to form [42]. Based on this study, the implant + oral BHA group was found to be effective in increasing VEGF expression on day 14 compared to the implant + oral calcium lactate group based on the IRS value of VEGF expression.

Osteocalcin is a non-collagenous protein that has an important role in the mineralization process and calcium ion homeostasis [43,44]. The results of the examination of osteocalcin expression in this study were all groups experienced an increase in osteocalcin expression since day 14. This is in line with the research of Jafary et al. [45], which states that an increase in osteocalcin can be detected at week 2, which is day 14. Osteocalcin can be detected in mature callus [4]. Mature callus begins to form from the repair phase to the remodeling phase [20]. Osteoblast maturation was demonstrated by increasing the value of osteocalcin expression [43]. Osteocalcin plays an important role in regulating mineral nucleation by binding to hydroxyapatite. Based on the study, the value of osteocalcin expression on days 14 and 28 showed a higher value of osteocalcin expression in the implant + oral BHA group than the implant + oral calcium lactate group. This is because BHA increases the proliferation of hydroxyapatite by stimulating mesenchymal cells, resulting in faster hard callus formation [18].

These results were also similar to the results of measuring BALP levels, where the group receiving oral BHA had higher BALP levels than the oral calcium lactate group. Bone alkaline phosphatase (BALP) is a metalloenzyme that is produced when osteoblast cells work [46]. High expression of BALP indicates better activity where BALP is useful in the synthesis of collagen fibers and in bone mineralization. On the 7th day, BALP levels were higher than the 14th and 28th days which could be due to the process of differentiation of osteoblasts. This is also in accordance with the theory which states that bone alkaline phosphatase (BALP) is a metalloenzyme that is produced when osteoblast cells work [46,47]. Osteoblasts will express collagen, which plays a role in the process of bone mineralization to a form soft callus. This is also in accordance with the statement of Einhorn and Gerstenfeld [2], and on the 7th day it enters the proliferative or endochondral phase where the soft callus begins to form. Soft callus growth also occurs because the gelatin contained in the implant is degraded to form type 1 collagen which is the main constituent of bone [8]. On the 14th day BALP levels decreased compared to the 7th day; this is because at this stage the process of fibrous tissue formation has occurred and soft callus maturation has begun, which has occurred on the 7th day [2]. On day 14, the mean number of osteoblasts also decreased as, according to the matrix maturation phase, osteoblasts lost their function and differentiated into osteoid in the bone [48]. On day 28, BALP also decreased, which could be due to the near-complete fracture healing process [49]. Birmingham et al. [50] also stated that the differentiation of osteoblast cells occurred at the beginning of day 5 to 14, after which there was a decrease in BALP expression. In addition, the results of the examination of higher blood calcium levels in the implant + oral BHA group compared to the implant + oral calcium lactate group will also increase the proliferation and/or chemotaxis of osteoblasts, thereby stimulating bone formation significantly.

Calcium lactate is known to be highly soluble and has high bioavailability; however, hydroxyapatite, which is generally considered insoluble, shows absorption values one-fourth to one-third as good as the most soluble preparations [51]. Hydroxyapatite raises blood calcium levels less than calcium carbonate and calcium citrate. This indicates that hydroxyapatite is more effective in entering bone cells [18]. In addition, in vitro examination of bone cell cultures showed that the organic part of OHC contains factors that influence osteoblast proliferation [13]. This causes the bone growth of the implant + oral BHA group to be better than the implant + oral calcium lactate group. BHA also has a carbonate group that is naturally present in human bone and that can be found in synthetic biomimetic hydroxyapatites. Carbonated hydroxyapatite is known to increase protein adsorption and increase adhesion, proliferation, and osteogenic differentiation of mesenchymal stem cell tissue. In addition, carbonated hydroxyapatite is also known to increase the proliferation and differentiation of osteoblasts, thereby increasing bone matrix synthesis [36]. The same thing happened in the study by Castelo et al. [29], based on histological observations of bone using a fluorescent microscope showed that oral administration of ossein hydroxyapatite could increase bone formation compared to the group given hydroxyapatite alone or with calcium carbonate. In addition, this is influenced by the absorption to the bioavailability of the two supplements given. It is known that the absorption of microcrystalline hydroxyapatite compound (MCHC) is about 42.5% [19], and it is higher than the absorption of calcium lactate which is around 25% [24].The results showed that the bioavailability of calcium lactate in rats was 8.9 + 1.4% [52]. Meanwhile, hydroxyapatite is reported to have better bioavailability than calcium carbonate [14], and as good as or better than calcium gluconate [53].

## 5. Conclusions

BHA supplements, given orally together with BHA–GEL pellet implants, showed faster healing of bone defects compared to oral calcium lactate with BHA–GEL pellet implants.

## Figures and Tables

**Figure 1 polymers-14-04812-f001:**
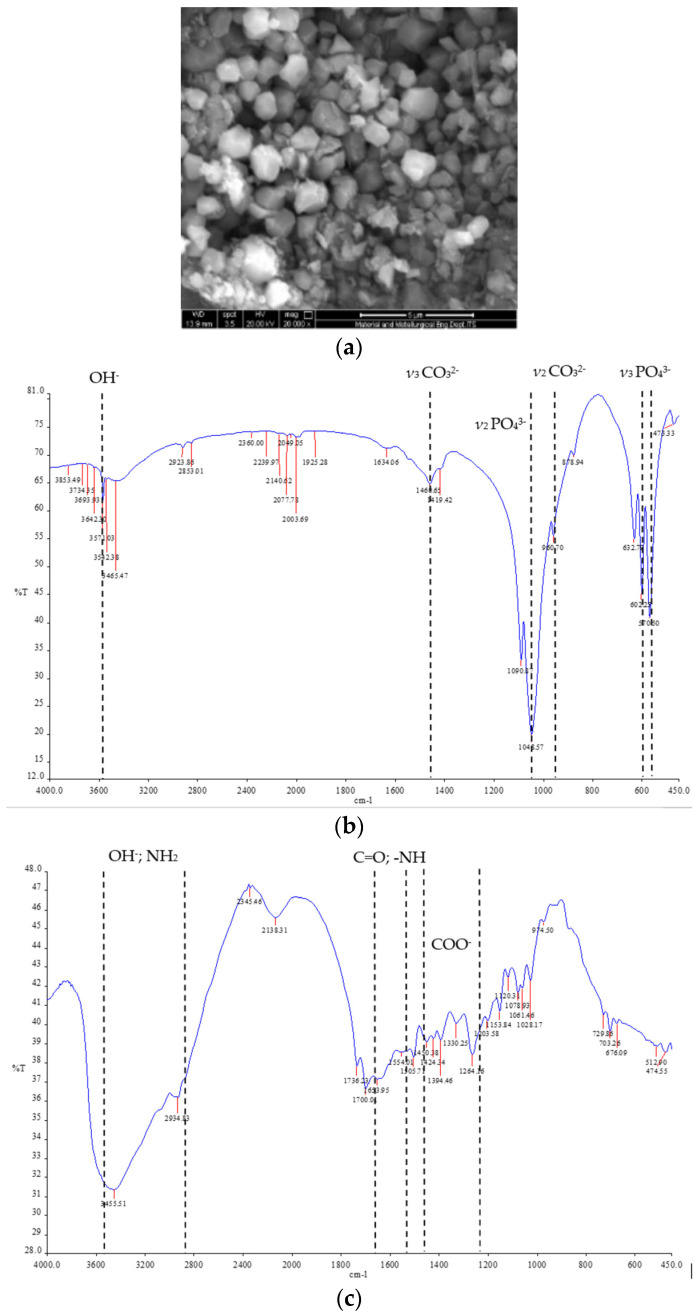
(**a**) Representative of SEM Scanning Result of BHA–GEL Pellet; (**b**) FT-IR Spectra Profile of BHA (**c**) FT-IR Sepctra Profile of Gelatin; (**d**) FT-IR Spectra Profile of BHA–GEL Pellet.

**Figure 2 polymers-14-04812-f002:**
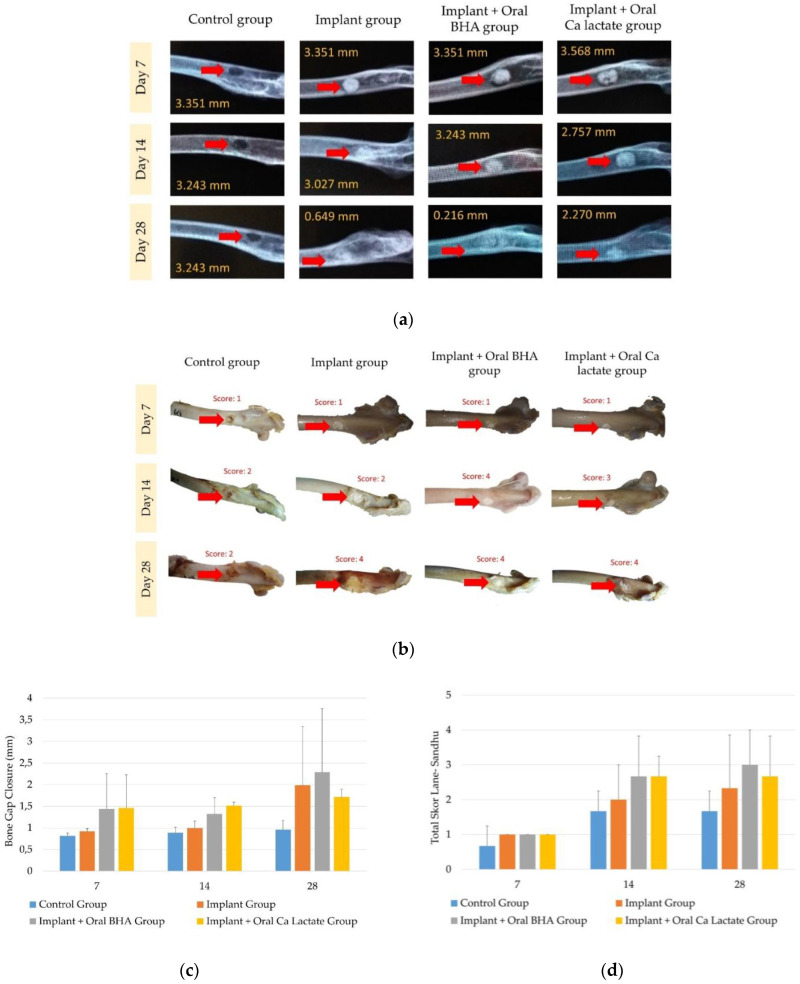
(**a**) X-ray radiology results of rabbit femur (1024 × 1024 pixel); (**b**) rabbit femur bone macroscopic; (**c**) result of measurement of bone cleft closure; (**d**) calculation results with Lane–Sandhu Scoring. The red arrow in subfigures (**a**,**b**) indicates the location of the defect. The implanted group showed accelerated bone growth in the defect area.

**Figure 3 polymers-14-04812-f003:**
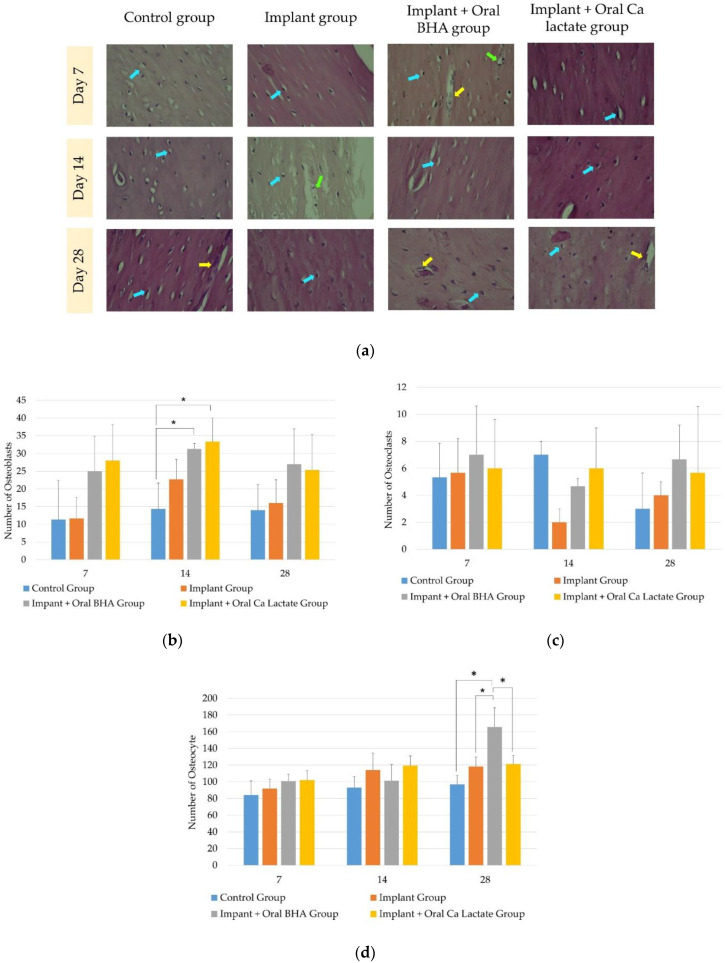
(**a**) Observation of bone cells with *Hematoxylin-Eosin* staining, magnification 400×. Osteoblast (yellow arrows), osteoclast (green arrows), and osteocyte (blue arrows); (**b**) observation of the number of osteoblast cells; (**c**) observation of the number of osteoclast cells; (**d**) observation of the number of osteocyte cells. The sign (*) indicates *p* < 0.05 with one-way ANOVA post hoc LSD test.

**Figure 4 polymers-14-04812-f004:**
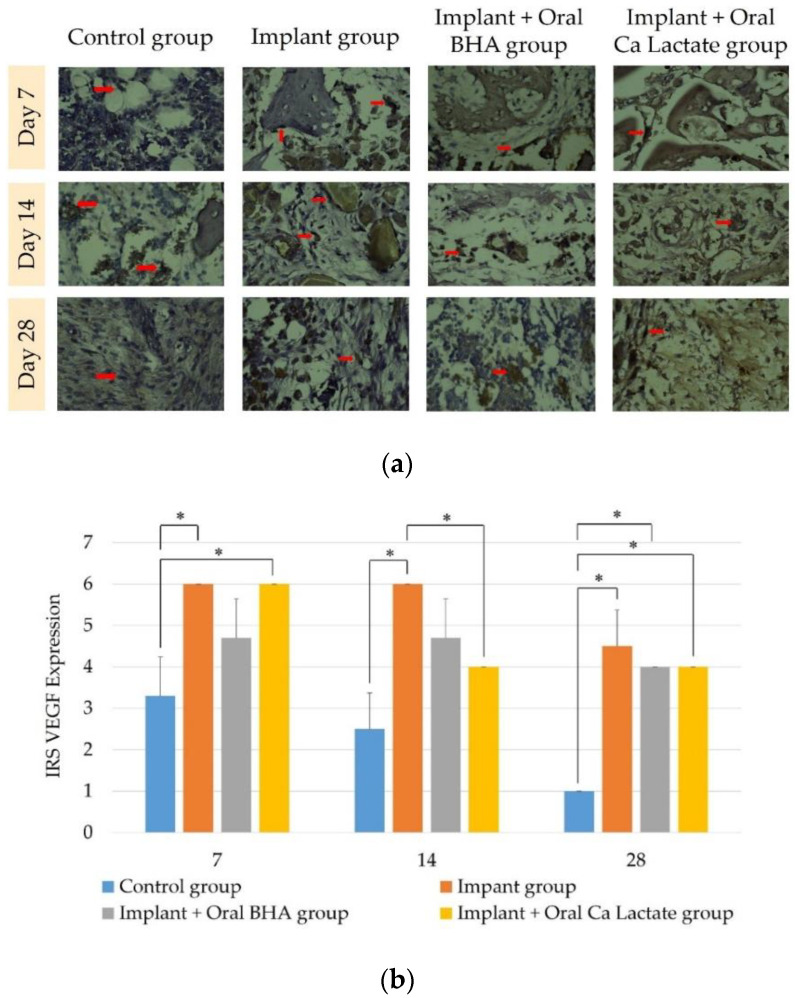
(**a**) Immunohistochemistry of VEGF expression (400× magnification). VEGF-positive cells are indicated by brown osteoblasts (red arrows); (**b**) differences in IRS values of VEGF expression, each bar graph represents IRS ± SD. The sign (*) indicates *p* < 0.05 with the Mann–Whitney Test.

**Figure 5 polymers-14-04812-f005:**
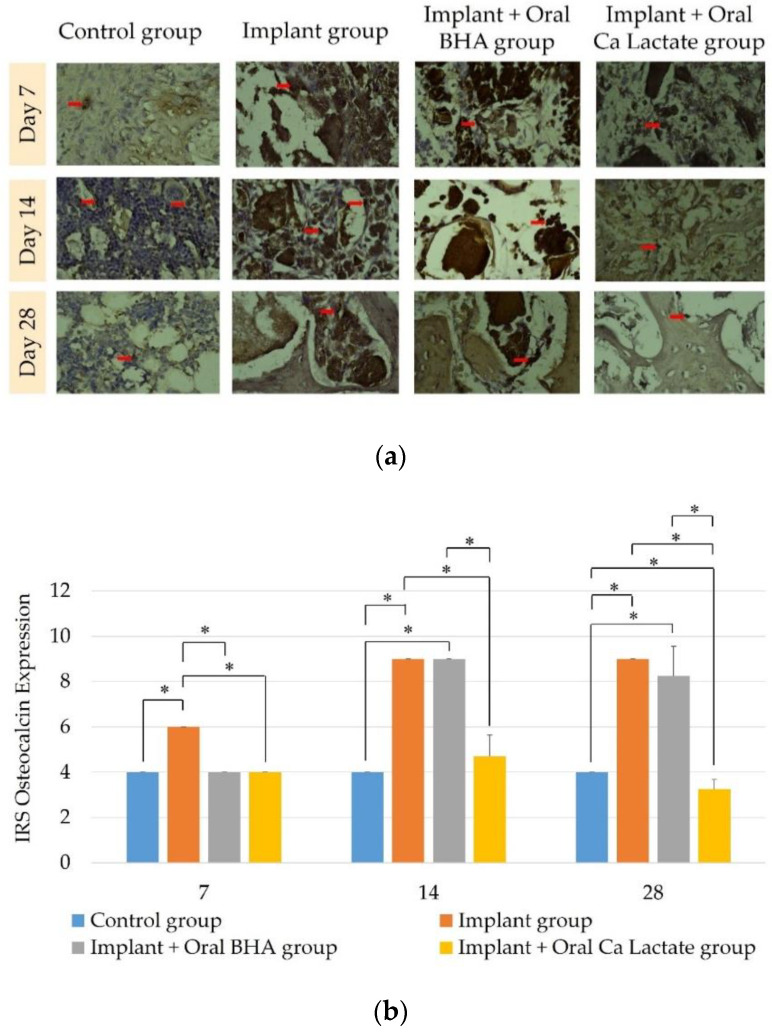
(**a**) Immunohistochemistry of osteocalcin expression (400× magnification). Osteocalcin-positive cells are indicated by brown osteoblasts (red arrows); (**b**) differences in IRS values of osteocalcin expression, each bar graph representing IRS ± SD. The sign (*) indicates *p* < 0.05 with the Mann–Whitney Test.

**Figure 6 polymers-14-04812-f006:**
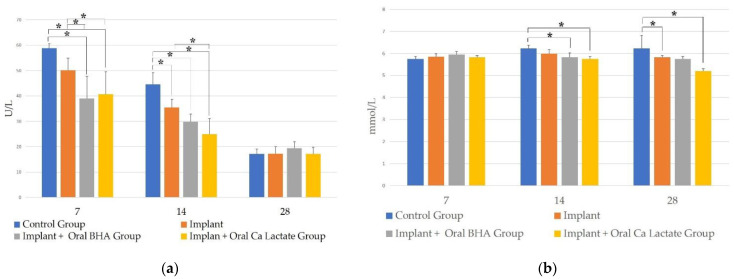
(**a**) BALP level examination results; (**b**) results of examination of calcium levels sign (*) indicates *p* < 0.05 in post hoc LSD test.

**Table 1 polymers-14-04812-t001:** Lane–Sandhu Scoring Criteria [27].

Criteria	Score	Characteristics
No callus	0	no callus tissue, fracture line clear
Minimal callus	1	25% callus tissue, fracture line still clearly visible
Callus evident but healing incomplete	2	50% callus tissue, fracture line blurred
Callus evident with stability expected	3	75% callus tissue, fracture line barely visible
Complete healing with bone remodeling	4	100% callus tissue, no remaining fracture line visible

## Data Availability

Not applicable.

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
