# Peer review of "Acceleration of Bone Fracture Healing through the Use of Bovine Hydroxyapatite or Calcium Lactate Oral and Implant Bovine Hydroxyapatite–Gelatin on Bone Defect Animal Model"

_polymers, 2022, doi:10.3390/polym14224812_

Round 1

Reviewer 1 Report

The results are interesting, and the data are statistically treated. The presented results are based on in vivo experiments and on histological and radiology observations. This should be interesting for a tissue engineering oriented journal however submission in Polymers should also combine the in vivo results with material’s characterization. Characterization of the prepared implants is totaly missed. The authors should present data showing at least  XRD and , SEM characterization of the implants and raw materials. Of course, other techniques such as FTIR, TGA should be also preferable if it is possible. To my opinion the manuscript should be published after major revision.

Minor comments

Page 52: Is the tricalcium phosphate b-tcp ? If yes add the term b-tricalcium phosphate

Author Response

We would like to express our thanks to you for all your suggestions that are needed to be revised to fulfil the requirements to be accepted in Polymer. We hope our responses to your comments meet your expectations. Please kindly let us know if you evaluate that this revision is proper enough to be accepted for publication or need further revision. We want to apologize about the miss words or sentences in our manuscript.

Reviewer #1

The results are interesting, and the data are statistically treated. The presented results are based on in vivo experiments and on histological and radiology observations. This should be interesting for a tissue engineering oriented journal however submission in Polymers should also combine the in vivo results with material’s characterization. Characterization of the prepared implants is totaly missed. The authors should present data showing at least  XRD and , SEM characterization of the implants and raw materials. Of course, other techniques such as FTIR, TGA should be also preferable if it is possible. To my opinion the manuscript should be published after major revision

Response to the general comments of reviewer #1: We would like to express our thanks to you for all your suggestions that are needed to be revised to fulfil the requirements to be accepted in Polymer. We hope our responses to your comments meet your expectations. Please kindly let us know if you evaluate that this revision is proper enough to be accepted for publication or need further revision. We want to apologize about the miss words or sentences in our manuscript. The characterization data of the pellets are being in our observation. The limited revision time made us meet any difficulties to fullfil your comment regarding to our characterization data.

Reviewer 2 Report

Dear authors, I have reviewed the manuscript entitled "Acceleration of bone fracture healing through the used of bovine hydroxyapatite or calcium lactate oral and implant bovine hydroxyapatite–gelatin on bone defect animal model " and I consider it acceptation for publication after minor revision.

My opinions and questions are as follows:

(1) Line 83: I disagree with your following statement: “BHA is known to have carbonate substitutions like that of human hydroxyapatite and not found in other synthetic hydroxyapatite synthetic [15].”

For example, you can find carbonated hydroxyapatite in the following papers:

Y. Wang, S. Von Euw, F.M. Fernandes, S. Cassaignon, M. Selmane, G. Laurent, G. Pehau-Arnaudet, C. Coelho, L. Bonhomme-Coury, M.-M. Giraud-Guille, F. Babonneau, T. Azaïs, N. Nassif, Water-mediated structuring of bone apatite, Nat. Mater. 12 (2013) 1144–1153, https://doi.org/10.1038/nmat3787.

Olivier F, Rochet N, Delpeux-Ouldriane S, et al. Strontium incorporation into biomimetic carbonated calcium-deficient hydroxyapatite coated carbon cloth: biocompatibility with human primary osteoblasts. Mater Sci Eng C. 2020;116:111192. doi:10.1016/j. msec.2020.111192.

I suggest adding these papers and changing your sentence. For example: “BHA is known to have carbonate substitutions like that of human hydroxyapatite and that can be found in synthetic biomimetic hydroxyapatites”.

(2) Line 193: Can you add images to illustrate the different scores?

(3) Line 279:  Can you add scales in Figure 1a? Can you give the resolution of this analysis? A µ-CT analysis would have been appropriate.

Figure 1b, 1c, 2b, 2c, 2d, 3b, …. Please enlarge all captions.

How did you calculate “bone gap closure”? You can add the explanations in "materials and methods".

(4) Line 300: Please put the correspondence between the colors of arrows and the type of cells in the caption.  

(5) Line 392: Did you measure the diameters? If so, can you provide them?

(6) Line 487: As with my first point, I disagree with the end of that sentence. You can find synthetic hydroxyapatite with carbonate groups such as most of biomimetic hydroxyapatites.

However, you are right to highlight the carbonate groups. Please change your sentence.

Author Response

We would like to express our thanks to you for all your suggestions that are needed to be revised to fulfil the requirements to be accepted in Polymer. We hope our responses to your comments meet your expectations. Please kindly let us know if you evaluate that this revision is proper enough to be accepted for publication or need further revision. We want to apologize about the miss words or sentences in our manuscript. Please see the attachment

Round 2

Reviewer 1 Report

My comments have been not answered at all in authors’ responses and in the revised manuscript

Author Response

Point-by point Response to the Reviewer’s comments:

Reviewer #1

The results are interesting, and the data are statistically treated. The presented results are based on in vivo experiments and on histological and radiology observations. This should be interesting for a tissue engineering oriented journal however submission in Polymers should also combine the in vivo results with material’s characterization. Characterization of the prepared implants is totaly missed. The authors should present data showing at least  XRD and , SEM characterization of the implants and raw materials. Of course, other techniques such as FTIR, TGA should be also preferable if it is possible. To my opinion the manuscript should be published after major revision

Response to the general comments of reviewer #1: We would like to express our thanks to you for all your suggestions that are needed to be revised to fulfil the requirements to be accepted in Polymer. We hope our responses to your comments meet your expectations. Please kindly let us know if you evaluate that this revision is proper enough to be accepted for publication or need further revision. We want to apologize about the miss words or sentences in our manuscript. The characterization data of the pellets are being in our observation. The limited revision time made us meet any difficulties to fullfil your comment regarding to our characterization data.

  1. Page 52: Is the tricalcium phosphate b-tcp ? If yes add the term b-tricalcium phosphate

Response to the Q1: Thank you for the critical question and suggestion. That's right, that's β-tricalcium phosphate and we have added in revised version of our manuscript (Page 2 Line 54)

Round 3

Reviewer 1 Report

I cannot see replies on my comments in the revised manucsript

Author Response

Point-by point Response to the Reviewer’s comments:

Reviewer #1

  1. Page 52: Is the tricalcium phosphate b-tcp ? If yes add the term b-tricalcium phosphate

Response to the Q1: Thank you for the critical question and suggestion. That's right, that's β-tricalcium phosphate and we have added in revised version of our manuscript (Page 2 Line 54)

  1. In order to have some physicochemical characterization of their implants the authors added FTIR graphs and a SEM image. Although other techniques are missed, the manuscript can be marginally accepted in a materials-oriented journal like “Polymers” after minor revision.

Minor comments

Add in lines 409-410 a number showing the mean size +/- SD  of the particles observed in SEM image

Response to the Q1: Thank you for the suggestion. We haved added the particle size mean in the text (Page 6 Line 276 and Page 13 Line 412)

Please kindly let us know if you that this revision needs further revision. Please find the reviewer's reply which attached here. We would be very happy if our work could be accepted for publication in internationally renowned Polymer

Thanks in advance,

Round 4

Reviewer 1 Report

In order to have some physicochemical characterization of their implants the authors added FTIR graphs  and a SEM image. Although other techniques are missed, the manuscript can be marginally accepted in a materials-oriented journal like “Polymers” after minor revision.

Minor comments

Add in lines 409-410 a number showing the mean size +/- SD  of the particles observed in SEM image

Author Response

Please kindly let us know if you that this revision needs further revision. Please find the reviewer's reply which attached here. We would be very happy if our work could be accepted for publication in internationally renowned Polymer

Thanks in advance,

Point-by point Response to the Reviewer’s comments:

Reviewer #1

  1. Page 52: Is the tricalcium phosphate b-tcp ? If yes add the term b-tricalcium phosphate

Response to the Q1: Thank you for the critical question and suggestion. That's right, that's β-tricalcium phosphate and we have added in revised version of our manuscript (Page 2 Line 54)

  1. In order to have some physicochemical characterization of their implants the authors added FTIR graphs and a SEM image. Although other techniques are missed, the manuscript can be marginally accepted in a materials-oriented journal like “Polymers” after minor revision.

Minor comments

Add in lines 409-410 a number showing the mean size +/- SD  of the particles observed in SEM image

Response to the Q1: Thank you for the suggestion. We haved added the particle size mean in the text (Page 6 Line 276 and Page 13 Line 412)